# Student Ratings: Skin in the Game and the Three-Body Problem

Charles Dziuban *, Patsy Moskal * , Annette Reiner, Adysen Cohen and Christina Carassas

Research Initiative for Teaching Effectiveness, Division of Digital Learning, University of Central Florida, Orlando, FL 32816, USA; annette.reiner@ucf.edu (A.R.); christina.carassas@ucf.edu (C.C.)
* Correspondence: charles.dziuban@ucf.edu (C.D.); patsy.moskal@ucf.edu (P.M.)

**Abstract:** To capture the student voice, university researchers examined the high-stakes Student Perception of Instruction form, administered online to students each semester, allowing them anonymous feedback on their courses. A total of 2,171,565 observations were analyzed for all courses each semester from fall 2017 through fall 2022. The results indicated that 68% of students responded identically to each of the protocol's 9 Likert scale items, essentially straight-lining their rating of instruction and casting doubt on the validity of their engagement with the process. Student responses by various University demographics are presented. We discuss the potential influences of students' reactions and present a possible model for effective teaching and evaluation.

**Keywords:** student ratings; student voice; student perception of instruction; higher education



## 1. Introduction

An ongoing concern in higher education is how to include the student voice in teaching. Most professional educators agree that doing so will improve educational effectiveness, better accommodate our diverse student population, and show that universities can respond to rapid societal changes. At the current time, the student voice primarily comes through two channels. The first is traditional and has been in place for almost a century [1]. In this approach, students provide feedback about their learning experience at the end of their courses using a rating scale instrument. Customarily, this process is formalized and controlled by a unit designated by the university administration. In theory, it has three functions: formative feedback for instructors, summative information for faculty evaluation, and lending credibility to the student voice.

However, it is no secret that the system has broken down for several reasons—one focus of this article. Students tell us they feel like robots rating every course but never seeing any tangible impact, so what is the point? They have no skin in the game because they perceive that their opinions do not impact change in the instructional practice. A second issue with this approach involves the usefulness of the data for any kind of valid faculty evaluation [2].

This led to the second "channel" for the student voice: an alternative, informal, uncontrolled, and virtual student evaluation of their courses and instructors. Students make their opinions available worldwide through sites such as ratemyprofessor.com, YouTube, X (formerly known as Twitter), Facebook, Instagram, TikTok, and Reddit. This "wild west" student evaluation happens in other spaces as well: fraternity and sorority houses, individual chats and text messages, businesses, and other places where students gather virtually or face to face. Faculty reputations are created in the alternative evaluation universe and spread like parasite memes, as Dawkins calls them in "The Selfish Gene" [3]. The reality is that this channel for student feedback continues to challenge the formal systems developed by universities as it is further reaching than the on-campus "form".

*1.1. Skin in the Game*

In the introduction, we used the term "skin in the game", indicating that students have no real investment in end-of-course ratings—and, for that matter, university faculty and administrators may not either. The term originated in the betting industry, where if a horse you own is in a race, you have skin in the game. The notion gained traction, referring to situations where individuals have a stake in the success or failure of a project or relationship, causing them to be personally invested in their actions and decisions.

The idea found widespread application in business and many aspects of society as a way to ensure that people assume responsibility and face the consequences [4–9]. In higher education, students assume more responsibility when they are actively engaged in their learning process, knowing that their efforts directly impact their futures. They gain a deeper understanding of the subject matter and develop critical thinking and problem-solving skills that allow them to apply their learning in real-world situations, preparing them for success beyond the classroom. Students who overcome obstacles become what Taleb [10] calls antifragile, developing strength from changing circumstances and building a foundation for lifelong learning. Similarly, educators who are committed to their students' success will make every effort to provide quality education and create a nurturing and supportive network that results in prepared and motivated graduates.

"Skin in the game" creates an atmosphere of accountability and ethical behavior in organizational leadership. However, its absence can lead to disastrous outcomes, as exemplified by the 2008 financial crash. McGhee [11] explains what happened when banks bypassed any responsibility for their subprime lending practices:

> The loans are called subprime because they're designed to be sold to borrowers who have lower than-prime credit scores. That's the idea, but it wasn't the practice. An analysis conducted for the Wall Street Journal in 2007 showed that the majority of subprime loans were going to people who could have qualified for less expensive prime loans. So, if the loans weren't defined by the borrowers' credit scores, what did subprime loans all have in common? They had higher interest rates and fees, meaning they were more profitable for the lender, and because we're talking about five- and six-figure mortgage debt, those higher rates meant massively higher debt burdens for the borrower". (p. 69)

> Never mind that most of the predatory loans we were talking about weren't intended to help people purchase homes, but rather, were draining equity from existing homeowners. (p. 89)

> Wall Street brokers even came up with a lighthearted acronym to describe this kind of hot-potato investment scheme: IBGYBG, for 'I'll be gone, you'll be gone.' If someone gets burned, it won't be us. (p. 92)

This is an example of what can happen when institutions feel free to exploit the underclasses, believing they are impervious to the consequences of their behavior. The irony of the situation was that as long as housing prices continued to rise, the scheme worked; but, as soon as they began to fall, the system collapsed.

Student course ratings appear to have minimal skin in the game for the constituencies involved. From a student's perspective, the time and effort taken to complete course evaluations has no effect on the course or the professor. In most cases, instructors only see their ratings after the course is completed. There is an absence of psychological contracts between faculty and students about how an evaluation system will function. The financial rewards for faculty are at most minimal, so their ratings have virtually no impact on salary increases. All parties concerned are suspect of the metrics provided by these data, and university administrators are skittish about high-stake decisions based on the evaluations. University bodies like the faculty senate are quick to criticize the system but have little to offer in the way of alternatives. In most instances, more comprehensive approaches are so labor-intensive that the opportunity costs are prohibitive. Often, in universities, the responsibility for redesigning the faculty evaluation procedures falls to dotted line units

such as the faculty center that only have the authority to make recommendations. At the moment, faculty ratings by students resemble Catch-22 [12]. Nobody wants to be evaluated in the current system because the results are suspect, but if you do not evaluate courses, you are not committed to teaching effectiveness, so you keep using a system you do not trust. Yossarian would be proud.

### 1.2. The Three-Body Problem

Another issue in this study hinges on student ratings in the context of the three-body problem: predicting the motion of three bodies under common gravitational forces. Although appearing unrelated to student ratings, the issue clarifies understanding students' evaluation because parallels between the two typify the complex dynamics of instructional effectiveness in higher education [13–17]. The challenge for both physics and education lies in their mutual complexity and the difficulty of obtaining exact solutions because of uncertainty and unpredictability [18,19]. Three fundamental issues underlie the problem.

1.  Interaction complexity: The culture of higher education involves complex interactions among students, instructors, curriculum, and course content.
2.  Inherent unpredictability: In both contexts (physics and education), the result is a long-term chaotic pattern. The interaction of student ratings with such things as teaching style, student engagement, overall experience, and individual student dispositions typifies a complex system. Addressing this unpredictability is key to understanding the student voice.
3.  Positive feedback loops: Student ratings experienced a sustained positive feedback loop reinforcing the system. We have been doing this for years, so change is hard, and really, the ratings do tell us something. Faulkner [20] is reputed to have said "a fellow is more afraid of the trouble he might have than he ever is of the trouble he's already got". Early typewriters, for example, tended to jam their keys—especially fast typists. To solve the problem, the letters QWERTY were placed on the upper left corner of the keyboard to separate the most used letters. This slowed the typists and reduced the jamming. Of course, typists became familiar with the arrangement and grew more proficient, thereby increasing efficiency. As new companies manufactured typewriters, there was no point in another keyboard arrangement because QWERTY was in place and universally used. Typists were trained in that system, creating an autocatalytic positive feedback loop that dictated the production of keyboards that has endured for 150 years. Student ratings underwent a similar positive reinforcement cycle, causing them to endure for almost 100 years.

The Three-Body Problem analogy to student ratings presents an open-ended challenge: no general solution exists because initial starting points are best guesses. This task before us is to devise entrepreneurial approaches that lead to satisfactory solutions [21–23]. This requires innovation, creativity, critical thinking, and trial and error. Embracing this uncertainty, ambiguity, and ambivalence can result in a sustainable and effective system for the assessment of teaching and learning from the student's perspective.

## 2. What the Literature Says: An Alternative Approach

### 2.1. A Seismic Shift in the Literature Review Paradigm

Examining Table 1, the number of articles about student evaluation of teaching identified by seven different platforms confirms a daunting problem for reviewing the literature on any topic. The internet, the cloud, electronic journals, blogs, videos, and a host of social media platforms have created literature bases that defy systematic analysis. Because of their constant churn and the discrepancies in numbers, traditional literature reviews have become increasingly difficult. A raft of other problems exists as well: overwhelming size, vague and overlapping classifications, mislabeling, excessive redundancy, inaccurate identification, and search tediousness.

**Table 1.** An emergent property representation of student rating literature.

| Author(s) | Summary |
|---|---|
| Google Scholar | 507,000 |
| Academic Search Premier | 21,623 |
| Pro Quest | 173,249 |
| JSTOR | 63,288 |
| ERIC | 61,776 |
| World Wide Science | 687,670 |
| Web of Science | 34,836 |

However, in recent months, artificial intelligence (AI), or more accurately, large language models, have lifted the concept of AI out of its doldrums, where it languished for years. Procedures such as neural networks, classification and regression trees, and nearest neighbor methods have enabled platforms such as ChatGPT to process huge amounts of information bits almost instantly, giving the impression of semantic thought. Floridi [24], however, offers a caution about that misconception in his article "AI As Agency Without Intelligence: On ChatGPT, Large Language Models, and Other Generative Models". He frames it this way:

> They do not think, reason, or understand; they are not a step towards any sci-fi AI; and they have nothing to do with the cognitive processes present in the animal world and, above all, in the human brain and mind, to manage semantic contents successfully [25]. However, with the staggering growth of available data, quantity and speed of calculation, and ever-better algorithms, they can do statistically—that is, working on the formal structure, and not on the meaning of the text they process—what we do semantically, even if in ways (ours) that neuroscience has only begun to explore. Their abilities are extraordinary, as even the most skeptical must admit. (pp. 1–2)

> The exercise is no longer to make summaries without using ChatGPT, but to teach how to use the right prompts (the question or request that generates the text. (p. 2)

These generative models are finding application in situations ranging from, but by no means bounded by, medical diagnosis to literary critique and analysis. Therefore, it is not surprising that these platforms have found their way into reviews of literature. For instance, Kabudi et al. [26] demonstrated an approach to using generative AI where specified apriori categories had the platform select initial literature sets and then apply multiple criteria to identify the most relevant subsets. The platform then "examined" those resources and placed clusters of articles into reasonably homogenous groups by aligning them with a strategic labeling process. This allowed the investigators to evaluate and organize their review. That platform accomplished what no group could do in a professional lifetime. Several authors cited the potential of these generative large-language AI platforms:

- Makes searching for relevant articles much faster [23,27–32]
- Has the ability to write entire summaries within seconds [30,33–35]
- Extremely effective for the editing process: checking grammar, creating citations, making an outline, etc. [27,36,37]
- Can help synthesize the chosen articles [29,31,34]

*2.2. A Blended Approach*

Table 2 represents the results of an incomplete traditional review summary of the literature conducted by the authors, but instead of a narrative, the results are presented in tabular form and classified (by the authors) under unifying subcategories. This typifies a folksonomy where the topic headings emerge in a self-organizing pattern characteristic of complex systems. Next, the authors independently identified subcategories under each organizing heading, then, as a group, negotiated the consensus. Based on that negotiation,

they designed a graphic visualization of the literature that provides a structural framework and connections to individual research papers. This addresses the micro−macro problem where reviewing individual articles does not necessarily produce a model that identifies important patterns. However, this semantic approach is labor-intensive and rests on the assumption that the sample of articles selected is representative of the body of literature. Figures 1–4 present the visual result of this analysis (micro to macro) with the author-identified categories.

**Table 2.** Student rating literature citations from several platforms.

| Resource | "Student Evaluation of Teaching" |
|---|---|
| **Course Modality, Level, and Content** | |
| Royal, K.D., & Stockdale, M.R. [38] | Students are more critical of professors teaching quantitative courses |
| Dziuban, C., & Moskal, P. [39] | Students do not consider course modality when completing evaluations |
| Glazier, R.A., & Harris, H.S. [40] | Students rate professors positively based on their teaching type regardless of course modality |
| Samuel, M. L. [41] | Students rated instructors in flipped classroom settings significantly higher |
| Liao, S., Griswold, W., Porter, L. [42] | Peer instruction with small groups consistently received higher ratings than larger, lecture-based classes |
| Capa-Aydin, Y. [43] | Students rated the in-class course much higher than the online course |
| Uttl, B., Smibert, D. [44] | Students rated quantitative courses significantly lower than non-quantitative courses |
| Brocato, B.R., Bonanno, A., & Ulbig, S. [45] | Instructors teaching online courses received lower ratings from students; Female instructors were rated higher |
| Filak, V.F., & Nicolini, K.M. [46] | Students were less satisfied with their online courses than face-to-face courses |
| Sellnow-Richmond, D., Strawser, M. G., & Sellnow, D.D. [47] | Online and hybrid students value flexibility but wish for more interaction and lecture-based teaching |
| Lowenthal, P., Bauer, C., Chen, K. [48] | Students rate online courses lower than face-to-face courses; graduate students are more critical of online course instructors; students rated tenured and tenure-track faculty lower than adjuncts |
| Yen, S.-C., Lo, Y., Lee, A., & Enriquez, J.M. [49] | Students in online, face-to-face, and blended formats were equally satisfied with their learning outcomes |
| He, W., Holton, A., Farkas, G., & Warschauer, M. [50] | Ratings on flipped instruction vs. traditional lectures were not significantly different |
| Mather, M., & Sarkans, A. [51] | Online students enjoy flexibility and convenience but want more timely feedback and interaction |
| Turner, K.M., Hatton, D., & Theresa, M. [52] | Online classes are rated lower than in-person; undergraduate students are more critical; larger classes receive lower ratings; classes with heavy workloads receive lower ratings |
| Peterson, D.J. [53] | Students in flipped classes rated course/professor higher than students in traditional lecture-based courses |
| **Student Factors (Decision, Perception)** | |
| Dziuban, C., Moskal, P., Kramer, L. & Thompson, J. [54] | As student ambivalence increases, so does the number of elements they use to evaluate their courses |
| Kornell, N., & Hausman, H. [55] | Students are unaware of what constitutes "good teaching" and just evaluate based on their class |
| Ernst, D. [56] | Students consider many factors when making the decision to fill out evaluations |
| Dziuban, C., Moskal, P., Thompson, J., Kramer, L., DeCantis, G., & Hermsdorfer, A. [57] | Understanding psychological contracts plays an important role in student satisfaction |

**Table 2.** *Cont.*

| Resource | "Student Evaluation of Teaching" |
|---|---|
| Griffin, B. [58] | Autonomy in courses leads to higher satisfaction and ratings |
| Richmond, A., Berglund, M., Epelbaum, V., Klein, E. [59] | Higher student ratings are based on the rapport between student and teacher, level of engagement, and personality of the professor |
| Scherer, R., Gustafsson, J.E. [60] | Students who achieved more in the course gave higher ratings |
| Gündüz, N. and Fokoué, F. [61] | A strong association exists between a student's seriousness/dedication and the ratings they assign to the course/professor; Identified zero variance responses |
| Bassett, J., Cleveland, A., Acorn, D., Nix, M., & Snyder, T. [62] | The majority of students only occasionally put significant effort into their rating responses |
| **Instructor Factors (Role, Perception, and Impact)** | |
| Mandouit, L. [63] | Student feedback is an important tool and powerful stimulus for instructor reflection |
| Wang, M.C., Dziuban, C.D., Cook, I.J., & Moskal, P.D. [64] | Instructor interest in their students' learning resulted in excellent ratings; low respect exhibited by instructors resulted in poor ratings overall |
| Golding, C., & Adam, L. [65] | Provides strategies for teachers to take student ratings into account when improving their teaching for future courses |
| Floden, J. [66] | Student feedback is perceived positively by university teachers, has a large impact on their teaching, and helps improve courses |
| Badur, B. and Mardikyan, S. [67] | Teachers with well-prepared courses, positive attitudes, and part-time professors consistently received higher ratings |
| Kim, L.E., & MacCann, C. [68] | Instructor personality impacts a student's evaluation of their teaching |
| Foster, M. [69]) | Professors addressed by their first name receive higher ratings than those who go by their title/last name |
| **Bias and Validity Concerns (gender and background in university decisions, based on a student's personal success)** | |
| Mengel, F., Sauermann, J., & Zolitz, U. [70] | Female professors receive lower ratings compared to their male counterparts |
| Stark, P.B., & Freishtat, R. [71] | Ratings may be reliable but are not necessarily valid/accurate; universities should abandon using student evaluations as the primary factor for promotion and tenure decisions |
| Heffernan, T. [72] | Abusive and rude comments common toward female professors and professors from minority backgrounds |
| Tejeiro, R., Whitelock-Wainwright, A., Perez, A., Urbina-Garcia, M.A. [73] | Students who received higher grades and are academically successful provide higher course evaluations |
| Stott, P. [74] | Students with poor grades are likely to rate their online instructors poorly |
| Esarey, J. & Valdes, N. [75] | Imprecision in the relationship between student evaluations and instructor quality |
| Kogan, V., Genetin, B., Chen, J., and Kalish, A. [76] | Students with better grades are more satisfied and leave higher ratings; not ideal to use evals for important decisions |
| Boring, A., Ottoboni, K., & Stark, P.B. [77] | Student evaluations are biased against female instructors |
| Flaherty, C. [78] | Evaluations tend to be biased against women; need to explore gender bias and tenure decisions |
| Flaherty, C. [79] | Major university decisions are in the hands of students who may be biased against their professors who are female or from racial minorities |
| Flaherty, C. [80] | Validity concerns due to grade satisfaction play a major role in how students evaluate |
| Flaherty, C. [81] | Student evaluations contain measurement bias and equity bias |
| Genetin, B., Chen, J., Kogan, V., & Kalish, A. [82] | Gender and racially implicit bias language on student evaluations need to be changed so students can still share concerns but not at the expense of their instructors |

**Table 2.** *Cont.*

| Resource | "Student Evaluation of Teaching" |
|---|---|
| Stroebe, W. [83] | Grade inflation may be due to student evaluations being used for determining major university decisions |
| Ray, B., Babb, J., & Wooten, C.A. [84] | Women instructors are held to a higher standard and have to work harder to be seen as competent |
| Goos, M., & Salomons, A. [85] | A low student response rate creates positive selection bias, meaning true evaluation scores may be lower |
| Boring, A., Ottoboni, K., & Stark, P. [86] | Female instructors receive lower scores than male instructors; students who expect to receive a higher grade are more likely to give higher ratings |
| Mitchell, K.M., & Martin, J. [87] | Considerable discrimination against female instructors in student ratings |
| Hornstein, H.A. [88] | Validity concerns regarding student evaluations are common |
| Buser, W., Batz-Barbarich, C., & Hayter, J. [89] | Female instructors rated significantly lower than male instructors; a student's expected grade strongly predicts their ratings |
| Chatman, J., Sharps, D., Mishra, S., Kray, L., & North, M. [90] | Even if a female instructor has similar performance as their male counterparts, they are still rated significantly lower |

Subsequently, however, Table 2 was submitted to ChatGPT where the authors asked the platform to identify four categories under each major heading. That result is also contained in Figures 1–4, showing a close (not exact) correspondence to the authors' work. This macro result helps validate the organizing structure of the research literature in student ratings of their courses from a combination of human cognition and machine learning—perhaps a shift in the way forward for capturing research findings that resonate with the digital age.

This review of student ratings in higher education is organized by four fundamental factors: course modality, student and instructor context, and validity. Each one plays a significant role in shaping student perceptions and experiences. Considering them from a macroperspective offers a comprehensive understanding of the issues. Course modality sets the stage for understanding the student's learning experience. Student and instructor contexts represent two personal components of course evaluation. However, conducting a review of the literature must embrace validity elements that influence student responses.

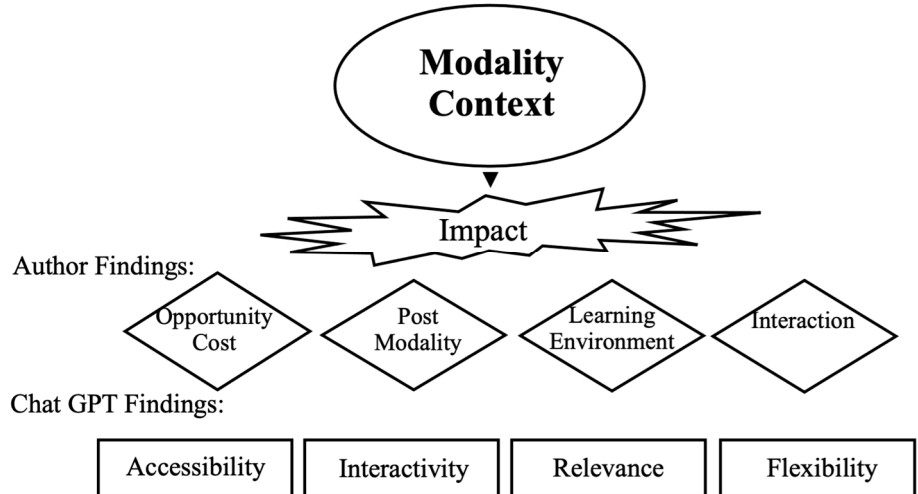

**Figure 1.** The learning arrangement construct.

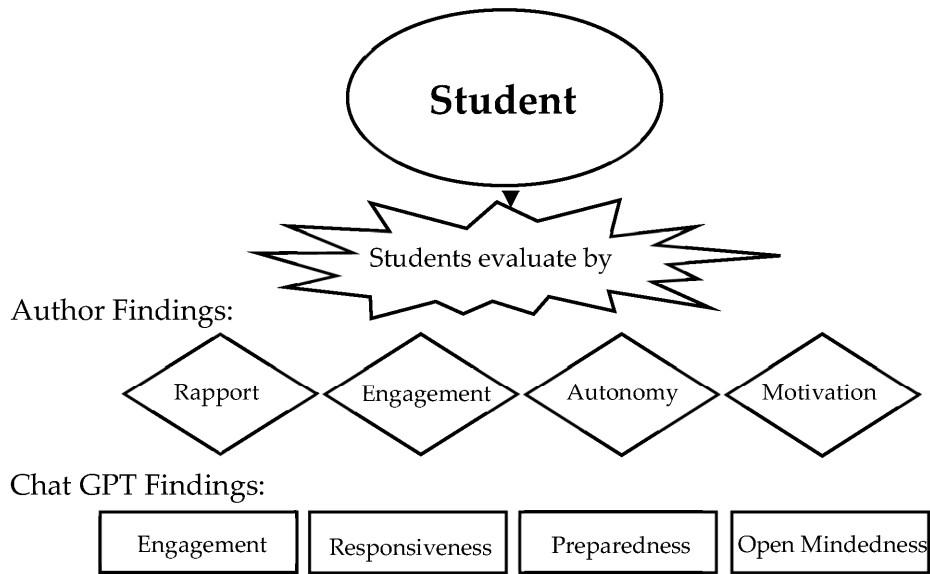

**Figure 2.** The student involvement construct.

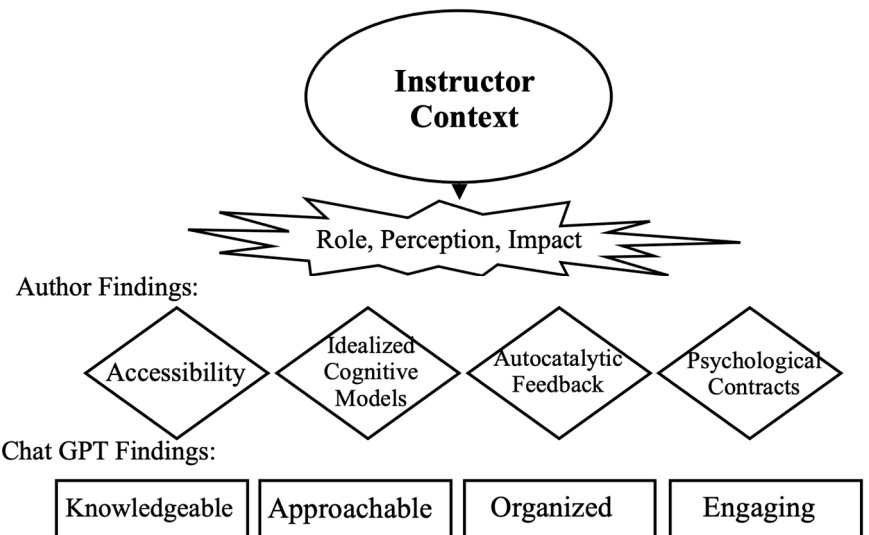

**Figure 3.** The teaching environment construct.

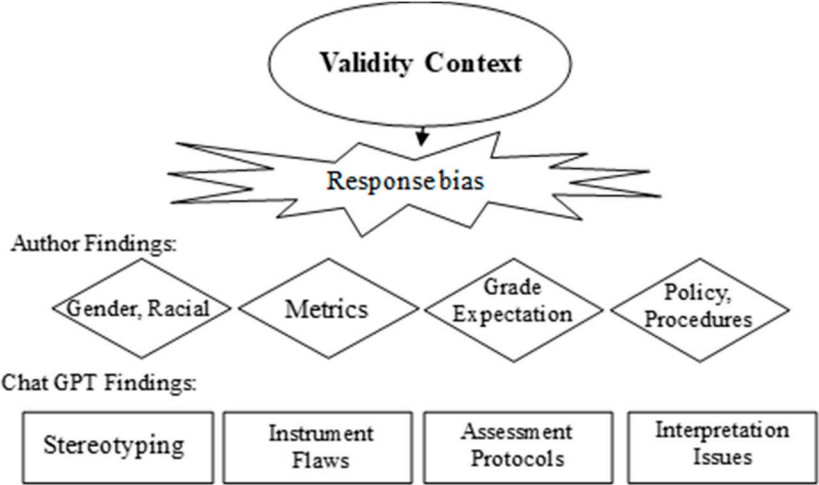

**Figure 4.** The measure quality construct.

Incorporating technology and utilizing approaches such as human semantic analysis and AI-based analysis like GPT enhances the process of analyzing the overwhelming number of articles. In this world of evolving technological innovations, conducting a valid review of the literature requires a multifaceted approach that considers the interplay of many factors enhanced by augmenting categories. By analyzing these factors in their interactive complexity, educators, administrators, and researchers can gain a more universal understanding of the variables affecting course evaluations.

## 3. What the Data Show

### 3.1. The Data Collection Procedures

The end-of-course Student Perception of Instruction at the University of Central Florida was the data source for this study (Appendix A). The rating scale has been re-designed and modified several times, with the current version resulting from a series of faculty, student, and administration groups working collaboratively to improve the process. The rating section comprises nine Likert items and two open-ended responses. The final version was approved by the faculty senate and was first administered in spring 2013. In addition to the instrument redevelopment, the committees addressed the strengths and weaknesses of the rating scale approach, recommending ethical use of the data for faculty evaluation and professional development. Student responses are anonymous, preventing the identification of any individual. Administration takes place online for all classes, irrespective of modality, managed by the university's information technology unit that provides summary results by course and makes the findings available to the faculty members, supplemented with departmental norms. Instructors and departments make individual determinations about data use, with some using it for promotion and tenure. The ratings are also used in some university faculty awards. The current study is based on the student responses to the instrument from the fall 2017 to the fall 2022 semesters and comprises 2,171,565 observations. Students are asked to respond to each item on a five-point Likert Scale (5 = excellent, 4 = very good, 3 = good, 2 = fair, 1 = poor). See Appendix A for the instrument.

### 3.2. The Data Analysis Plan

The original protocol called for an analysis of the results for the entire responding student group by computing a total score over the nine items and examining the data. Then, from a measurement perspective deriving indices of internal consistency (Alpha) and item analysis, including difficulty analogs and discrimination [91]. This was to be followed by determining the domain sampling properties of the data using the measure of sampling adequacy [92]. Subsequently, the investigators intended to determine distributional characteristics by computing the moments (central tendency, variability, skewness, and kurtosis). Upon establishing the psychometric adequacy of the data, the objective was to use the total scores as the criterion measure for the differential impact of course modality, college, department, course level, class decile, and pre, during, and post-COVID timeframes, avoiding statistical hypotheses testing because of excessive power. The plan was to assess the differences by computing effect sizes and obtain a consensus about their importance and impact on the instructor evaluation process.

### 3.3. An Unexpected Anomaly and The Results

The student rating process on university campuses is a good example of a complex system. Forester [93] cautions us that one can never predict how an intervention will ripple through a complex system for instance, moving the rating system online. Also, outcomes will be counterintuitive, and there will be side effects that must be accommodated. That is what happened in this study. Earlier, we indicated that we started by calculating the total scores. That is when the anomaly arose. We noted a disproportionate number of total scores that summed up to 45. For the nine-item instrument, the only way that could happen would be nine responses with ratings of five each. Therefore, this side effect atomized the

focus of the study by creating an emergence encountered in complex systems where the interactions are more meaningful than the individual components. Most likely, this will become a characteristic of contemporary educational and social research. This phenomenon was pointed out in an article by Gündüz and Fokoué [61], where they termed these patterns zero variance. We called this straight lining and followed up by checking the additional total scores of 36, 27, 18, and 9. Obviously, a total score of nine requires responses from all ones. The remaining total scores, 36, for instance, could indicate that a student selected all fours; however, there are multiple combinations of responses that would sum to that value and not indicate zero variance. Therefore, we examined that possibility as well. The result of that research in Table 3 shows that 68% of the over 2 million responses exhibited straight-lining responses. Table 4 shows the percentage of that behavior for each item in the rating scale. Although not 100% for other items (excluding 45 or 1), the percentages are very high. Table 5 shows that by far (70%) the straight-lining involved all 5s, with substantially smaller percentages for the other total scores.

**Table 3.** Percentage of students who responded identically (straight liners) on the SPI: 2017–2022.

|  | N | Percent |
|---|---|---|
| No | 695,528 | 32.0 |
| Yes | 1,476,037 | 68.0 |

**Table 4.** Percentage of students who responded identically (straight liners) for each item on the SPI: 2017–2022 based on total score.

| Total Score | 45 (5) N 1,034,022 | 36 (4) N 205,539 | 27 (3) N 190,327 | 18 (2) N 53,491 | 9 (1) N 39,456 |
|---|---|---|---|---|---|
| Organizing | 100 | 94.9 | 96.3 | 93.2 | 100 |
| Explaining | 100 | 94.6 | 96.3 | 92.9 | 100 |
| Communicating | 100 | 93.9 | 95.5 | 91.1 | 100 |
| Respect and concern | 100 | 96.3 | 97.0 | 94.1 | 100 |
| Interest | 100 | 95.1 | 96.2 | 92.7 | 100 |
| Environment | 100 | 93.3 | 94.9 | 90.5 | 100 |
| Feedback | 100 | 95.1 | 96.3 | 92.7 | 100 |
| Achieve | 100 | 92.9 | 94.5 | 89.9 | 100 |
| Overall effectiveness | 100 | 90.8 | 93.4 | 87.7 | 100 |

**Table 5.** Frequency and percentage of students who responded identically (straight liners) on the SPI: 2017–2022.

| Score | N | % Straight Line |
|---|---|---|
| All 5s | 1,034,022 | 70.1% |
| All 4s | 182,800 | 12.4% |
| All 3s | 174,828 | 11.8% |
| All 2s | 44,931 | 3.0% |
| All 1s | 39,456 | 2.7% |

*3.4. A Change in Plans*

These findings caused the investigators to abandon the total score as an outcome measure and change to a binary variable—whether students straight-lined or not. Examining Table 3 shows that only 32% of students responded to the items somewhat independently. This could indicate a more considered approach to evaluating their courses, although this is an assumption that has not been verified. But at least they are not straight-lining. This creates a contingency analysis for two categorical variables. Therefore, the relationship index changed to the lambda coefficient [94,95] that assesses the strength of association between two categorical variables, with 1 indicating a perfect relationship and 0 indicating

complete independence. The results of that analysis are presented in Tables 6–11. The lambda value for each contingency table was zero, indicating that none of the independent variables had any impact on whether students straight-lined or not. The behavior was ubiquitous across all aspects of the university. Students straight-lined (zero variance) the rating scale at a ratio of 2 to 1.

**Table 6.** Percentage of students by course modality who responded identically (straight liners) on the SPI: 2017–2022.

| Modality | N | Straight Line % |
|---|---|---|
| Reduced seat time mixed mode (M) | 207,046 | 67.3% |
| Face-to-face (P) | 951,287 | 65.8% |
| Initial reduced face-to-face (R) | 25,308 | 62.9% |
| Reduced seat time, active learning (RA) | 32,479 | 63.9% |
| Limited attendance (RS) | 62,210 | 69.4% |
| Video streamed with classroom attendance (RV) | 16,279 | 63.4% |
| Video streamed (V) | 51,243 | 65.6% |
| Synchronous "live" video (V1) | 165,981 | 68.8% |
| Online (WW) | 659,732 | 71.7% |

**Table 7.** Percentage of students by college who responded identically (straight liners) on the SPI: 2017–2022.

| College | N | Straight Line % |
|---|---|---|
| Arts and Humanities | 247,173 | 65.5% |
| Business | 258,828 | 66.8% |
| Community Innovation & Education | 172,679 | 73.1% |
| Education | 22,676 | 66.7% |
| Engineering & Computer Science | 254,170 | 62.7% |
| Health & Public Affairs | 51,808 | 75.0% |
| Health Professions & Sciences | 111,450 | 76.8% |
| Medicine | 77,309 | 70.3% |
| Nursing | 61,555 | 74.8% |
| Sciences | 699,005 | 67.0% |
| Graduate Studies | 1934 | 68.9% |
| Nicholson School of Communication & Media | 44,519 | 64.5% |
| Rosen School of Hospitality Management | 75,937 | 69.6% |
| School of Optics | 3183 | 57.8% |
| The Burnett Honors College | 2805 | 57.8% |
| Undergraduate Studies | 14,840 | 74.4% |

**Table 8.** Percentage of students by department * who responded identically (straight liners) on the SPI: 2017–2022.

| Department | N | Straight Line % |
|---|---|---|
| Army ROTC | 2067 | 88.5% |
| Communication | 32,960 | 65.8% |
| Criminal Justice | 40,790 | 76.1% |
| Economics | 41,010 | 55.1% |
| Electrical & Computer Engineering | 29,272 | 58.8% |
| School of Kinesiology & Physical Therapy | 23,951 | 78.3% |
| Marketing | 28,054 | 71.3% |
| Mechanical & Aerospace Engineering | 75,152 | 68.3% |
| Nicholson School of Communication & Media | 51,167 | 65.5% |
| Tourism, Events, and Attractions | 28,359 | 68.6% |

* A randomly selected subset.

**Table 9.** Percentage of students by class size decile who responded identically (straight liners) on the SPI: 2017–2022.

| Class Size Decile | N | Straight Line % |
|---|---|---|
| 1.00 | 221,597 | 67.3% |
| 2.00 | 222,634 | 66.8% |
| 3.00 | 223,338 | 67.3% |
| 4.00 | 220,884 | 66.9% |
| 5.00 | 214,709 | 69.3% |
| 6.00 | 213,737 | 70.3% |
| 7.00 | 222,869 | 69.9% |
| 8.00 | 200,981 | 66.1% |
| 9.00 | 213,532 | 66.5% |
| 10.00 | 216,284 | 69.2% |

**Table 10.** Percentage of students by course level who responded identically (straight liners) on the SPI: 2017–2022.

| Course Level | N | Straight Line % |
|---|---|---|
| Lower Undergrad | 734,318 | 66.3% |
| Upper Undergrad | 1,277,164 | 69.8% |
| Graduate | 156,300 | 60.9% |
| Total | 2,167,782 | |

**Table 11.** Percentage of students pre- and during COVID, who responded identically (straight liners) on the SPI: 2017–2022.

| | N | Straight Line % |
|---|---|---|
| Pre-COVID | 874,945 | 66% |
| During COVID | 653,662 | 70% |
| Post-COVID | 642,958 | 69% |

## 4. What Does This Mean?

### 4.1. The Three-Body Problem and a Possible Explanation

Obviously, this is an unexpected and concerning finding. Apparently, two-thirds of students (1,476,037) are not engaged meaningfully in the evaluation of their courses. They demonstrate that they have no skin in the game with the straight-line response pattern. Perhaps they view that the opportunity costs of thoughtful responses far outweigh the added value of the process. In focus groups, they reinforce their opinions that they do not see the impact of their responses, although these data can be very high stakes for faculty members. Students express their feelings on social media but seem reticent to express them in the formalized system. However, there is a possible alternate explanation for this behavior. The fact that the predominance of the straight-lining occurs at the excellent level might indicate that this is a comprehensive evaluation of the course and instructor and that the students view item-by-item variable responses as contributing little added value to their end-of-course responses. This would have a significant impact on a comparative metric approach to his information. This is particularly concerning when one thinks about summarizing the data for colleges and departments when most of the students have bypassed the system. This has implications far beyond the hypothetical biases and impacts found in the research literature: modality, student context, instructor context, and validity. Those constructs simply do not apply if students are not engaged in any meaningful way. This is a conundrum. If they are not involved, why? Figure 5 presents a possible explanation cast in the context of the three-body problem. The figure posits the three driving forces in the problem, ambivalence characterized by simultaneous positive and negative feelings about rating their courses. Indifference—defined as being unconcerned or

uninvolved in a particular situation or towards a specific action. Ambiguity—the quality of being open to more than one interpretation or having multiple possible meanings. This occurs when something is unclear, uncertain, or can be understood in different ways, leading to confusion or difficulty in understanding its true intent or significance. The interaction of the three forces produces additional influences. Detached refers to being emotionally disengaged or impartial, often in a situation where meaningful involvement is expected. Apathetic describes a lack of interest, enthusiasm, or concern about something—the absence of motivation to engage in a particular situation or task. Indifference refers to being uncaring and showing little or no reaction towards the things happening around them. Equivocal refers to situations or requirements that can be interpreted in different ways, making it difficult to determine the underlying purpose behind them. This represents a complex pattern of interacting forces that, when considered as a system, hinders students in their attempts to evaluate their courses. With all these elements creating a positive reinforcement cycle, the optimal decision might be just to straight line the rating form.

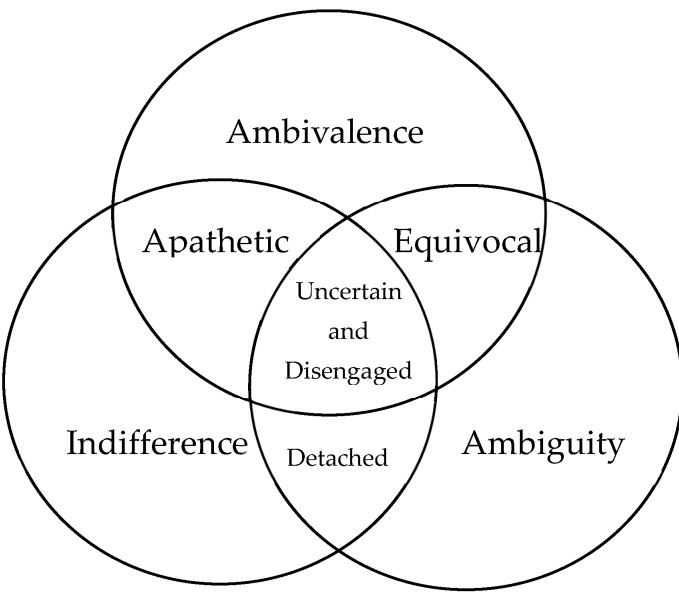

**Figure 5.** The Three-Body Problem and Student Disengagement.

The elements of the three-body problem are not unique to the evaluation of the course issue. They exist in many contexts: science, society, education, technology, humanities, history, politics, and medicine, just to mention a few. Additionally, these emotional and cognitive states are replete in contemporary and classic literature. For example, consider Table 12, which cites the protagonists in popular works, each one characterizing one of the dispositions in Figure 5.

**Table 12.** The Three-Body Problems in Literature.

| Component | Character | Book | Author |
| --- | --- | --- | --- |
| Ambivalence | Agnes | The Old Drift | Namwali Serpell |
| Indifference | Okonkwo | Things Fall Apart | Chinua Achebe |
| Ambiguity | Sethe | Beloved | Toni Morrison |
| Detached | Cora Randall | The Underground Railroad | Colson Whitehead |
| Equivocal | Ifemelu | Americanah | Chimamanda Ngozi Adichie |
| Apathetic | Bigger Thomas | Native Son | Richard Wright |
| Perplexed | David | Giovanni's Room | James Baldwin |

### 4.2. What If Common Sense Does Not Make Sense?

On the face of it, students' ratings of their courses appear to make sense because it can serve as an important feedback mechanism for educational institutions. However,

these assumptions seem flawed when most students are not actively participating in the process. Additionally, ratings can be influenced by personal biases or grievances rather than objective course evaluation. Students may lack the expertise to assess the effectiveness of pedagogical methods or curriculum design accurately. Despite their potential benefits, student rating systems should be viewed in the context of contemporary educational complexity. It may be that commonsense has led us astray.

Duncan Watts' [96] and Daniel Kahneman's [97] thinking offers insights into how student ratings can create biased, inaccurate, and misleading interpretations. Watts' work defining social networks is relevant, showing that course evaluations are not isolated events but are part of a larger network of interactions. The ratings are impacted by forces such as social connections, the instructor's reputation on social media, or commonly held attitudes. Watts' work on perception bias reinforces the argument that individual evaluations could be misleading because a small number of excessively positive or negative impressions may dominate the overall reaction to a course. He would contend that it is crucial to embrace a broader system of interactions and the diversity of approaches to develop a more comprehensive understanding of a course's effectiveness [96]. Kahneman's research on cognitive biases makes a strong case that the availability heuristic influences people's judgments. When they recall one specific positive or negative incident, that recollection will overly influence their general evaluation because an exceptionally enjoyable or frustrating experience will overshadow the overall experience. Additionally, the anchoring effect might impact students' ratings because when they contrast one course to another, an exceptional experience anchors their expectations, unfairly influencing their evaluation of their current course [97]. As suggested by our findings, social desirability bias might well impact how students rate their courses. They will be disposed to assign positive ratings, especially if they see it as the socially acceptable response while deferring on criticism to avoid potential conflicts or repercussions. Perhaps this is why we found 70% all 5 s and less than 3% all 1 s.

*4.3. An Evolving Context*

So many things have changed since a hundred years ago when educators believed that there would be value in having students rate their courses. At that time, there was only one face-to-face modality; the primary delivery method was the lecture, and the technology of choice was the chalkboard. However, instructional technologies began making their way into classrooms with the to-be-expected furor, but they persisted. Their impact is old news, and by now, the number of higher education course modalities in the digital environment has made the traditional concept of the class, what Susan Leigh Starr has termed a boundary object—strong enough to hold a community of practice together but weak in terms of definition in the larger community although strong in individual constituencies [98]. Without a unified and accepted class model, to what are students responding?

A second contextual issue is the increasing financial and educational inequity in our country. Current data show that if a student resides in the lowest economic quartile, then their chances of obtaining a college degree are eleven percent [99]—the odds against them are nine to one. These are terrible odds. These young people are living a life of what Mullainathan and Shafir [100] call scarcity, where their needs far exceed their resources, causing them to juggle so many things in their lives just to survive—adding college study to that list causes all the dominos to collapse and the optimal decision for them is to drop out with no chance of ever returning. The total accumulated college debt in the country is 1.7 trillion dollars [101]. This is staggering. If that were a gross domestic product, it would be the ninth-largest economy in the world. And it should surprise no one that most of that debt is carried by those in the lowest economic classes [102]. The cost of higher education in the United States denies access to so many. As a result, we are wasting millions of perfectly good minds simply because they do not have access to the resources necessary to succeed. Unfortunately, this inequity and bias have increased run-away decision-making by opaque

and non-transparent technologies with a built-in, programmed bias that makes important decisions about people and their lives. Consider this from O'Neal [103]:

> Nevertheless, many of these models encoded human prejudice, misunderstanding, and bias into the software systems that increasingly manage our lives. Like gods, these mathematical models were opaque, their workings invisible to all but the highest priests in their domain: mathematicians and computer scientists. (p. 3)
>
> OR
>
> Without feedback, however, a statistical engine can continue spinning out of faulty and damaging analysis while never learning from its mistakes. They define their own reality and use it to justify their results. This type of model is self-perpetuating, highly destructive—and very common. (p. 7)

In addition, there is a distinct college access wealth advantage in this country. A recent *New York Times* article showed that children from wealthy families have a far greater chance of getting into an elite university than their disadvantaged peers, even though their academic credentials are equivalent [104]. The evidence goes even further. Research shows that those affluent graduates have far better access to prestigious jobs simply because of the trailing wind of wealth advantage. Gumbel [105] states:

> Put another way, people from upper-middle-class origins have about 6.5 times the chance of landing an elite job compared to people from working-class backgrounds. Origins, in other words, remain strongly associated with destinations. (p. 13)
>
> OR
>
> As root a Bourdieusian lens insists that our class background is defined by our parents' stocks of three primary forms of capital: economic capital (wealth and income), cultural capital (educational credentials and the possession of legitimate knowledge, skills, and tastes), and social capital (valuable social connections and friendships). (p. 14)

The Supreme Court recently vacating affirmative action on university campuses caused a vehement backlash so much so that the department of justice launched an investigation into donation and legacy admissions, especially at elite institutions. Consider this quote from a New York Times article by Cochrane et al. [106]:

> With the end of race-based affirmative action, the practice of giving admissions preference to relatives of alumni is particularly under fire at the most elite institutions, given the outsized presence of their alumni in the nation's highest echelons of power. A new analysis of data from elite colleges published last week underscored how legacy admissions have effectively served as affirmative action for the privileged. Children of alumni, who are more likely to come from rich families, were nearly four times as likely to be admitted as other applicants with the same test scores. (para. 8)

This inequity is further reinforced by the recent admission to elite universities scandals [104]. All these events may seem far away from student rating of instruction, but they are not. Consider how underserved students would be equipped to rate their classes and instructors compared to their affluent classmates who inherit a strong sense of agency and entitlement at universities. Jack [107] discusses how first-time college students from underserved communities experience an entirely different institution:

> Some students discover, to their great consternation, that they are also responsible for deciphering a hidden curriculum that tests not just their intellectual chops but their ability to navigate the social world of an elite academic institution, where the rewards of such mastery are often larger and more durable than those that come from acing an exam. (p. 86)

How would you aggregate end-of-course rating data from these two distinct cohorts in a class, and how would you interpret what those data mean?

Finally, the COVID pandemic had and is having a dramatic impact on universities and public schools, where both were forced to not only keep the doors open with virtual education but also attempt to maintain quality. In the initial move to emergency remote instruction when the world locked down, the impact was devastating. The long-term effect is yet to be experienced, but we are already seeing signs of what is to come. A significant segment of the current generation is not including a college education in their post-secondary education plans [108]. Further, this generation is much less prepared for university work than most any other group in recent decades [109]. These contexts have a dramatic impact on how students perceive their higher education: how they experience it, how they react, and how they express their opinions.

### 4.4. An Idealized Cognitive Teaching Evaluation Model

Figure 6 presents our concept of an effective and supportive teaching evaluation system in contemporary universities. To be sure, this represents a seismic shift in higher education's culture, and for the moment is purely speculative. However, given the dysfunction of the current rating system, change might emerge through:

1.  Teaching First Commitment: Dedication to and valuing teaching excellence equally with other academic pursuits by recognizing the influence educators have on students.
2.  A Culture of Teaching Effectiveness: A shared commitment to continuous improvement in teaching methodologies, encouraging instructors to adapt according to student needs informed by the scholarship of teaching and learning.
3.  Comprehensive Formative Evaluation (excluding summative evaluation): Providing constructive, systematic feedback to instructors through formative assessments rather than using student evaluation for comparisons.
4.  Prototype Exemplary Teaching: Celebrating and learning from superior instructors who inspire and engage students, setting a benchmark for instructional excellence.
5.  Actionable Teaching Insights: Utilizing research-based insights and innovative teaching methods to bridge the gap between theory and practice.
6.  Evaluation-Grounded Feedback: Leveraging student ratings and other evaluation protocols to support professional development.

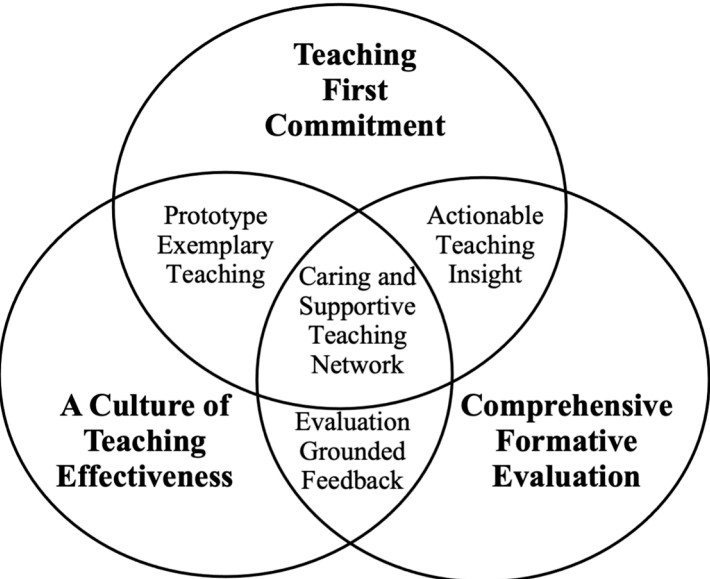

**Figure 6.** A Three-Body Possibility for Effective Teaching and Evaluation.

The interplay of these elements will establish a Caring and Supportive Teaching Network, fostering an educational community of practice that emphasizes cooperation

and promoting an environment for the personal and professional growth of all involved in teaching and learning. In such a university, a supportive teaching network would flourish, uniting faculty, students, and administration in a shared vision for academic excellence.

In keeping with the theme of this special issue, by asking if online instructional technology offers hope for higher education, the student evaluative voice becomes paramount. Online learning has transformed higher education by accommodating the lifestyles of individuals who are unable to displace themselves to attend on-campus courses typical in traditional education. This transformation has not only made higher education accessible to a broader demographic but has changed the learning landscape from an inward-focused to an outreach model. Digital learning removed barriers that once targeted higher education to a specific population. Now students, irrespective of location or family and work demands, can obtain further education in their own time, space, and motivation levels. As we noted previously, the COVID-19 pandemic demonstrated the value of online learning as a mechanism that was key to the continued functioning of American higher education. As campuses were forced to close their doors, this modality showcased the intrinsic value of being online as an effective, dependable, and flexible means of teaching and learning. By bridging geographical, educational, financial, and societal distances, the new modalities not only allowed American universities to survive the challenges of a pandemic but also simultaneously expanded their educational mission beyond the confines of traditional campuses. Our model, comprising the three primary elements, resonates with technologies that continue to advance as the learning landscape evolves. By harnessing the power of data analytics, fostering open communication, and embracing ongoing assessment, online instructors can create exemplary teaching experiences that empower students to reach their full potential with options such as:

- Content Personalization, enabling instructors to curate material that resonates with individual learners, creating a more engaging experience.
- Adaptive Learning that can dynamically adjust the difficulty and specificity of content and design assessments based on student performance, ensuring that each learner experiences effective learning trajectories.
- Automated Feedback, allowing for real-time generation of constructive information about student progress that enables timely positive learning interventions.
- Learning Analytics that assess knowledge acquisition patterns and create engagement metrics identifying areas of required improvement coupled with appropriate interventions.
- Natural Language Processing chatbots serving as virtual teaching assistants, answering students' questions, and providing guidance 24/7.
- Collaborative Platforms in which online classrooms can facilitate virtual group work, providing discussion prompts and analyzing group dynamics to encourage productive interaction.
- Automated Assessment that handles routine learning metrics, saving instructors time and effort and allowing them to focus more on personalized interactions with students and designing more complex evaluation methods.
- Sentiment Analysis might gauge student attitudes and engagement towards various aspects of the learning experience. This information can be used to tailor support and create a positive online learning environment.
- Large Language Generative AI Models that can enhance higher education by providing personalized learning experiences, customizing educational content, and providing real-time formative learning feedback with AI tutors.

Additionally, blended learning can leverage enhanced presentations by offering virtual office hours, thus enhancing student-centered pedagogy. Blended learning, as a combination of traditional face-to-face and online learning, has become transformative in higher education by maximizing the affordances of both modalities. Students can access course materials online, engage in interactive discussions, and collaborate with their classmates and instructors, establishing an effective support network. In the rapidly evolving educa-

tional environment, blended learning has emerged as a cornerstone of higher education, strengthening digital literacy and information fluency, and preparing students for the demands of our contemporary workforce. This learning innovation not only captures the best of both learning worlds but also supports diverse learning modes and will grow in importance in the coming years, preparing students to succeed in our knowledge-driven world [110].

As digital learning continues to evolve, its integration into traditional universities will become more seamless and impactful. However, it is essential to acknowledge that the successful integration of online learning into student evaluation of their courses requires careful planning, faculty training, and support from university administration. As learning continues to evolve, online education can become an effective platform for student evaluation by enabling a valid student voice in higher education.

In effective university environments, while research undoubtedly holds great significance for advancing the boundaries of human understanding, teaching emerges as an equally critical pillar deserving equivalent support and recognition. By creating a culture that values and supports both endeavors, universities can fulfill their transformative potential that is so vital in this technologically driven world, cultivating well-rounded scholars, both students and faculty empowering the coming generations with the knowledge and skills to make a meaningful impact on society. Of course, this change faces obstacles requiring formidable work, effort, and commitment—Muhammad and the mountain come to mind. Unfortunately, there is no Maxwell's demon to eliminate the friction. However, if we address the adjacent possible, the next reasonable first step, we will begin the journey. As Gwyn Thomas said, "the beauty is in the walking—we are betrayed by destinations". If this is quixotic, then bring on the windmills and let us continue our search for Dulcinea of Toboso.

**Author Contributions:** Conceptualization, C.D.; methodology, C.D.; software, C.D. and P.M.; validation, C.D. and P.M.; formal analysis, C.D. and P.M.; investigation, C.D.; resources, C.D. and P.M.; data curation, C.D. and P.M.; writing—original draft preparation, C.D.; writing—review and editing, C.D., P.M., A.R., A.C. and C.C.; visualization, C.D., P.M., A.R., A.C. and C.C.; supervision, C.D. and P.M.; project administration, C.D. and P.M. All authors have read and agreed to the published version of the manuscript.

**Funding:** This research received no external funding.

**Institutional Review Board Statement:** Not applicable.

**Informed Consent Statement:** Not applicable.

**Data Availability Statement:** UCF's Student Perception of Instruction data is available to university staff and faculty only or by request here: https://it.ucf.edu/our-services/test-scoring/student-perception-of-instruction/, accessed on 30 October 2023.

**Acknowledgments:** The authors would like to thank Tony Picciano for his careful editing and dedication to continuing to promote quality research through this special issue. We would also like to thank and acknowledge the tireless faculty who work to provide quality instruction for their students and whose service provides the context for our research.

**Conflicts of Interest:** The authors declare no conflict of interest.

## Appendix A

### Student Perception of Instruction

Instructions: Please answer each question based on your current class experience. You can provide additional information where indicated.

All responses are anonymous. Responses to these questions are important to help improve the course and how it is taught. Results may be used in personnel decisions. The results will be shared with the instructor after the semester is over.

**Please rate the instructor's effectiveness in the following areas:**

1.　　Organizing the course:

(a) Excellent (b) Very Good (c) Good (d) Fair (e) Poor

2.　　Explaining course requirements, grading criteria, and expectations:

(a) Excellent (b) Very Good (c) Good (d) Fair (e) Poor

3.　　Communicating ideas and/or information:

(a) Excellent (b) Very Good (c) Good (d) Fair (e) Poor

4.　　Showing respect and concern for students:

(a) Excellent (b) Very Good (c) Good (d) Fair (e) Poor

5.　　Stimulating interest in the course:

(a) Excellent (b) Very Good (c) Good (d) Fair (e) Poor

6.　　Creating an environment that helps students learn:

(a) Excellent (b) Very Good (c) Good (d) Fair (e) Poor

7.　　Giving useful feedback on course performance:

(a) Excellent (b) Very Good (c) Good (d) Fair (e) Poor

8.　　Helping students achieve course objectives:

(a) Excellent (b) Very Good (c) Good (d) Fair (e) Poor

9.　　Overall, the effectiveness of the instructor in this course was:

(a) Excellent (b) Very Good (c) Good (d) Fair e) Poor

10.　　What did you like best about the course and/or how the instructor taught it?

11.　　What suggestions do you have for improving the course and/or how the instructor taught it?

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
