# Peer review of "Student Ratings: Skin in the Game and the Three-Body Problem"

_education, doi:10.3390/educsci13111124_

Round 1

Reviewer 1 Report

Comments and Suggestions for Authors

The submitted manuscript discusses student ratings for a particularly implemented student evaluation of university courses. I found the paper not very focused, albeit fluently written. The fact that there is straight lining across all items referring to the best category could also mean that they choose the best category for a course but do not see value in providing detailed responses for every item. Hence, a straight lining approach could also be interpreted as a (from the student's perspective) efficient overall summary of a university course.

Author Response

The submitted manuscript discusses student ratings for a particularly implemented student evaluation of university courses. I found the paper not very focused [1], albeit fluently written. The fact that there is straight lining across all items referring to the best category could also mean that they choose the best category for a course but do not see value in providing detailed responses for every item. Hence, a straight lining approach could also be interpreted as a (from the student's perspective) efficient overall summary of a university course.[2]

[1] We appreciate your careful review of our manuscript and will wo do our best to respond to your comments. With respect to your suggestion about study focus we have added a statement that explains that the focus became atomized took on the emergent properties of a complex system.

[2] We have added your insightful suggestion about an alternate interpretation of the straight lining phenomenon as an indication of a generalized response to the course but a hesitancy to vary responses item by item. Excellent suggestion, thank you.

We hope this satisfies your concerns. We note that you marked four sections as must be improved without specific suggestions. We acknowledge that no manuscript is finished but simply abandoned. Every one can be improved in a never ending autocatalytic cycle. If you want to give us more specific recommendations, we will be happy to accommodate them. We tried to reflect a drastically complex process for converting data into usable information---a process that is become increasingly difficult. We do appreciate your comment about the fluency of our writing. Thanks again.

Reviewer 2 Report

Comments and Suggestions for Authors

Student Ratings: Skin in the Game and The Three Body Problem

Review comments:

To authors:

I thoroughly enjoyed reading this article. The authors bring to their narrative strong evidence, intelligence, wit, and wisdom on the topic of student ratings. While the research on student ratings has a long history, the authors bring new life to the problem of student ratings and engagement and the process of teaching and evaluation. I appreciate the level of thought and thoughtfulness that the authors devoted to this topic. The “problem” of lack of student engagement in course evaluation and straight lining is not solved by only a traditional, formalized system approach. Revamping is necessary. The authors bring hope to the idea that the “student voice” must be reconstituted and reshaped by a new model of innovative thinking and strategy towards engagement and institutional effectiveness. It requires a different level of institutional commitment that comprises a supportive teaching and instructional network designed for all stakeholders.  Students are less interested now in completing evaluations even when given the convenience of completing the “form” online (though perhaps still some institutions use paper forms?) – the transition has not led to positive transformation of the way students think about evaluation. I might be saying it a bit differently than the authors, but this is partly what I think I am getting from the paper. The traditional form and the process for which we express and analyze assessment of teaching and instruction has become a stumbling block and needs a complete transformation! Bravo!

Nothing in the paper requires substantive revision, but I do suggest parts that can be shortened up a bit. The paper is a bit long, but this is not a substantial problem. It is up to the authors to decide if they want to reduce parts, or not.

The Introduction is an excellent exposition of the problem and focus of the paper. The case is expertly made for “skin in the game”, from the student’s perspective. The three-body problem analogy is also a great fit: interaction complexity, inherent unpredictability, positive feedback loops – these are established so well in the paper. The QWERTY keyboard has endured a positive feedback loop for 150 years, perhaps so because it has also evolved in its physicality or modality in the way it is used. The traditional student rating system has not benefited from positive evolution from physical form to online form. Though it has moved from paper to digital (or online), the modality transition has not necessarily improved efficiency, and has not transformed the efficacy or perception of assessment.

About the second channel -- the authors make this very important statement: “The reality is that this channel for student feedback continues to challenge the formal systems developed by universities as it is farther reaching than the on-campus ‘form’.”

Perhaps we do need to investigate options to further channel student feedback beyond the traditional form or even the so called newer online rating system, such options as informal spaces, polling software, other alternatives that might motivate students to engage in the process. Something fun, something fun for the instructors, too.

Suggested Change: I have no suggested changes to the Introduction, but I do suggest the authors revisit some of these important statements (like the one quoted above from lines 45-46) in the conclusion of the paper, to bring out the future of the evaluation system and how it might function better in the future.

Section 2. What the Literature Says……

While I like this section, I almost feel it is a bit too involved in describing the process the authors used in organizing and finding patterns in the student rating literature. Table 2 is quite lengthy, and the figures seem sort of tacked on after the table. I do not think the figures as standalone are critical to the representation of the literature. While it is nice to show how ChatGPT could surely enhance the process for review generation and analysis of literature review, and presumably will improve its accuracy in time, the ChatGPT findings in this early stage do not seem to have a significant impact on identifying the constructs described in the analysis of student ratings. My suggestion would be to remove lines 162-199 and eliminate reference to ChatGPT and the figures in section 2.2 A Blended Approach.

Table 2 – would be nice to have the year of publication to follow the pattern of how the research has evolved over time.

Section 3. What the Data Show

I appreciate how “University of XXX” has devoted efforts to redesign a usable and short evaluation form.

The entire section is excellent. The results identifying straight liners are interesting, compelling, and staggering at the same time. It is amazing what these patterns show.

No suggested changes to this section.

Section 4. What Does This Mean?

4.1. I like this section and the figure. Could you please check if you intend Indifference to be mentioned twice – line 350 and line 358.

I suggest removing lines 368-374, including Table 12. It would condense things a bit and shorten the paper.

Section 4.2 What if Common Sense Doesn’t Make Sense.

Brilliant. No changes.

Section 4.3. An Evolving Context

Everything indicated is carefully crafted and thoughtfully regarded. However, this section is very long. I wonder if some of the text on financial and educational inequity could be reduced, the text between lines 418-490, could be reduced. Pick up at line 491, which by the way, is an awesome question – bring it out early!!!!!!

4.4. An Idealized Cognitive Teaching Evaluation Model

Brilliant. You give the reader much to think about.

Lines 532-577, our model, awesome comprehensive approach and visionary to say the least!

I’m wondering if content in lines 579-609 is a bit off-topic or at least maybe not as effective as a conclusion. I would like to see something that ties back to the notion of the skin in the game and creating future channels for the student voice. Revisit some of the early statements from the paper, bring them back and look into the future.

Overall, the paper is an excellent presentation. There are a few places where condensing might shorten the paper, but these are only suggested ideas. I will enter the decision “accept with minor revision” in case the authors wish to adjust the paper. I’m looking forward to seeing this paper published ASAP.

Author Response

We appreciate your obvious careful review of our manuscript and the many valuable insights you provided. These projects always need constant improvement, and we are painfully aware of that in our preparation of this piece. We hope that because you assigned yes to each category that you will be comfortable leaving the article intact. However, your insights paved the way for our next article on the student evaluative voice in higher education. Thank you.